# Neoadjuvant durvalumab plus radiation versus durvalumab alone in stages I–III non-small cell lung cancer: survival outcomes and molecular correlates of a randomized phase II trial

We previously reported the results of a randomized phase II trial (NCT02904954) in patients with early-stage non-small cell lung cancer (NSCLC) who were treated with either two preoperative cycles of the anti-PD-L1 antibody durvalumab alone or combined with immunomodulatory doses of stereotactic radiation (DRT). The trial met its primary endpoint of major pathological response, which was significantly higher following DRT with no new safety signals. Here, we report on the prespecified secondary endpoint of disease-free survival (DFS) regardless of treatment assignment and the pre-specified exploratory analysis of DFS in each arm of the trial. DFS at 2 and 3 years across patients in both arms of the trial were 73% (95% CI: 62.1–84.5) and 65% (95% CI: 52.5–76.9) respectively. For the exploratory endpoint of DFS in each arm of the trial, three-year DFS was 63% (95% CI: 46.0–80.4) in the durvalumab monotherapy arm compared to 67% (95% CI: 49.6–83.4) in the dual therapy arm. In addition, we report post hoc exploratory analysis of progression-free survival as well as molecular correlates of response and recurrence through high-plex immunophenotyping of sequentially collected peripheral blood and gene expression profiles from resected tumors in both treatment arms. Together, our results contribute to the evolving landscape of neoadjuvant treatment regimens for NSCLC and identify easily measurable potential biomarkers of response and recurrence.

The standard of care for medically fit patients presenting with early-stage non-small cell lung cancer (NSCLC) is complete surgical resection, either alone or in combination with adjuvant or neoadjuvant therapy. However, despite potentially curative resection and perioperative therapy, recurrence develops in 30–50% of patients, usually within 12–24 months, suggesting the persistence of micro-metastatic disease despite delivery of the best available therapy. In recent years, blockade of the PD-1/PD-L1 immune checkpoint has not only transformed the treatment of patients with advanced NSCLC but has also made significant inroads into the therapeutic landscape of patients with early-stage disease[1–14]. For example, a new standard of care for patients with completely resected stage II/IIIA NSCLC is adjuvant chemotherapy followed by the PD-L1 or PD-1 blocking antibodies, atezolizumab, and pembrolizumab, respectively[3,4]. In the neoadjuvant

✉e-mail: nkaltork@med.cornell.edu; bi2175@cumc.columbia.edu; temcgraw@med.cornell.edu

space, blockade of the PD-1/PD-L1 immune checkpoint combined with preoperative chemotherapy is also a new standard of care for patients with stages IB-IIIA NSCLC regardless of PD-L1 expression[6–8]. Despite these significant advances, improvements in survival are principally driven by complete pathological responders (20–25%). Further improvements in outcomes associated with immune checkpoint blockade may be accomplished by dual checkpoint inhibition[13,14] or by combining immune checkpoint blockade with potentially immuno-modulating agents such as low-dose or metronomic chemotherapy, targeted anticancer agents, or modulatory doses and fractionations of radiotherapy[15–22]. Radiotherapy enhances immune response through multiple proposed mechanisms, including induction of immunogenic cell death with release of neoantigens, upregulation of the MHC complex and enhanced antigen presentation, activation of dendritic cells and enhanced antigen cross-presentation, modulation of check-point expression and increasing T cell infiltration into the tumor[23]. We recently reported the results of a randomized phase II trial comparing neoadjuvant durvalumab alone with neoadjuvant durvalumab plus stereotactic radiotherapy (SBRT) in patients with early-stage NSCLC[24]. Major pathological response (MPR), the primary endpoint of the trial, was observed in two of 30 patients (6.7%) in the monotherapy group and 16 of 30 resected patients (53.3%) in the dual therapy group. To our knowledge, this was the first neoadjuvant trial in early-stage NSCLC testing the hypothesis that neoadjuvant SBRT delivered as three daily fractions of 8 Gy to the primary tumor, is safe and acts as a potent immunomodulator of the tumor microenvironment, thus enhancing the anti-tumor immune response associated with immune checkpoint blockade. Here we report disease-free survival (DFS), the secondary endpoint of the trial, in the intention-to-treat population regardless of treatment assignment as well as the exploratory endpoint of the difference in DFS between both arms of the trial. We also report the results of an unplanned post hoc analysis of progression-free survival (PFS) in both arms of the trial. Despite a higher proportion of patients achieving MPR after immune-radiotherapy, DFS was similar between arms in the intention to treat the population and among patients who had the planned surgical resection. However, in an unplanned post hoc analysis of PFS, we found that preoperative SBRT plus durvalumab was associated with a trend toward improved PFS at two and three years compared to the durvalumab alone arm. In a further post hoc analysis, high-plex immunophenotyping of sequentially collected peripheral blood from patients in both treatment arms identified a strong correlation between pretreatment circulating CD103+ T cells, which harbor tissue-resident memory (TRM)-like phenotype and MPR. We also found that among patients who do not achieve MPR in both treatment arms, an increase of immune pathway genes in resected tumors was correlated with freedom from recurrence.

## Results

Between 25 January 2017 and 15 September 2020, 60 patients who met eligibility criteria were enrolled and randomly assigned to either two cycles of durvalumab alone or two cycles of durvalumab plus three daily fractions of SBRT (8 Gy) delivered concurrently with the first cycle of durvalumab (Consort diagram; Supplementary Fig. S1). The baseline demographic, clinical, and pathological characteristics for all 60 randomized patients were previously reported and were well-balanced between arms[24]. All randomized patients received at least one dose of durvalumab and were therefore assessable for DFS. Twenty-six out of 30 patients in each arm underwent the planned surgical resection and were assessable for all survival endpoints. Surgical resection was not performed in four (13%) patients in the durvalumab alone group because of preoperative death in one patient (preoperative stroke), patient's wishes in one patient, and disease progression in two patients; one with pretreatment clinical stage IA who was found to have occult pleural implants at exploration and another patient with clinical stage IIIA who developed bone metastasis before the planned surgical procedure.

Similarly, surgical resection was not performed in four (13%) patients in the durvalumab plus radiotherapy group because of pre-operative death (cardiac event) in one patient and disease progression in three patients with pretreatment clinical stages IIA (aortic invasion), IB (pleural nodules), and IIIA (lung metastases). Demographic, clinical, and pathological characteristics of the 52 resected patients were well-balanced between arms (Table 1). As previously reported, major pathological response, the primary endpoint of the trial, was observed in 2/30 patients in the monotherapy arm (6.7%, 95% CI: 0.8% to 22.1%) and in 16/30 patients (53.3%, 95% CI: 34.3%–71.7%) in the dual therapy arm.

Adjuvant chemotherapy, either alone or combined with con-formal radiation according to the standard of care at that time, was offered to all patients and was declined by most patients who had MPR. Adjuvant cytotoxic therapy (chemotherapy ± radiation) was more frequently delivered in patients who had preoperative durvalumab (16/26) alone compared to patients who had preoperative SBRT and durvalumab (9/26), likely reflecting the higher proportion of patients with MPR in the radio-immunotherapy arm (Supplementary Table S1). In each arm of the trial, patients received a median of 4 cycles of chemotherapy. In the trial protocol, adjuvant durvalumab was optionally offered to patients who have either completed or declined adjuvant chemotherapy regardless of PD-L1 expression. Adjuvant durvalumab was given in 16/26 in the durvalumab alone arm and 18/26 patients in the SBRT plus durvalumab arm. The median number of cycles of adjuvant durvalumab was 4.5 in the monotherapy arm and 9.5 in the dual therapy arm.

### Disease-free survival

The key secondary endpoint of 2-year DFS, as prespecified in the protocol, was estimated in the intention to treat the cohort, including

**Table 1 | Baseline demographics and clinical characteristics**

| | Durvalumab | SBRT + durvalumab | *p*-values |
|---|---|---|---|
| | *n* = 26 | *n* = 26 | |
| Age, median (IQR), in years | 71 (63–76) | 69 (64–73) | 0.558[a] |
| **Gender** | | | |
| Male | 15 (58%) | 13 (50%) | 0.578[b] |
| Female | 11 (42%) | 13 (50%) | |
| **Race** | | | |
| White | 21 (81%) | 19 (73%) | 0.801[b] |
| African American | 3 (11%) | 4 (15%) | |
| Asian | 2 (8%) | 3 (11%) | |
| **ECOG performance status** | | | |
| 0 | 23 (88.5%) | 26 (100%) | 0.235[b] |
| 1 | 3 (11.5%) | 0 (0%) | |
| **Smoking** | | | |
| Never | 6 (23.0%) | 2 (7.7%) | 0.148[b] |
| Former/current | 20 (76.9%) | 24 (92%) | |
| **Clinical stage (7th ed)** | | | |
| Stage IA/IB | 2/7 (35%) | 1/6 (27%) | 0.455[b] |
| Stage IIA/IIB | 1/4 (19%) | 5/4 (35%) | |
| Stage IIIA | 12 (46%) | 10 (38%) | |
| **Clinical tumor size** | | | |
| Median, mm | 35.0 (30.5–53.8) | 45.0 (31.0–59.8) | 0.486[a] |
| **Histology** | | | |
| Adenocarcinoma | 13 (50.0%) | 16 (61.5%) | 0.506[b] |
| Squamous cell carcinoma | 10 (38.5%) | 9 (34.6%) | |
| Other | 3 (11.5%) | 1 (3.8%) | |

[a]Two-tailed Mann-Whitney test.
[b]Chi-squared test.

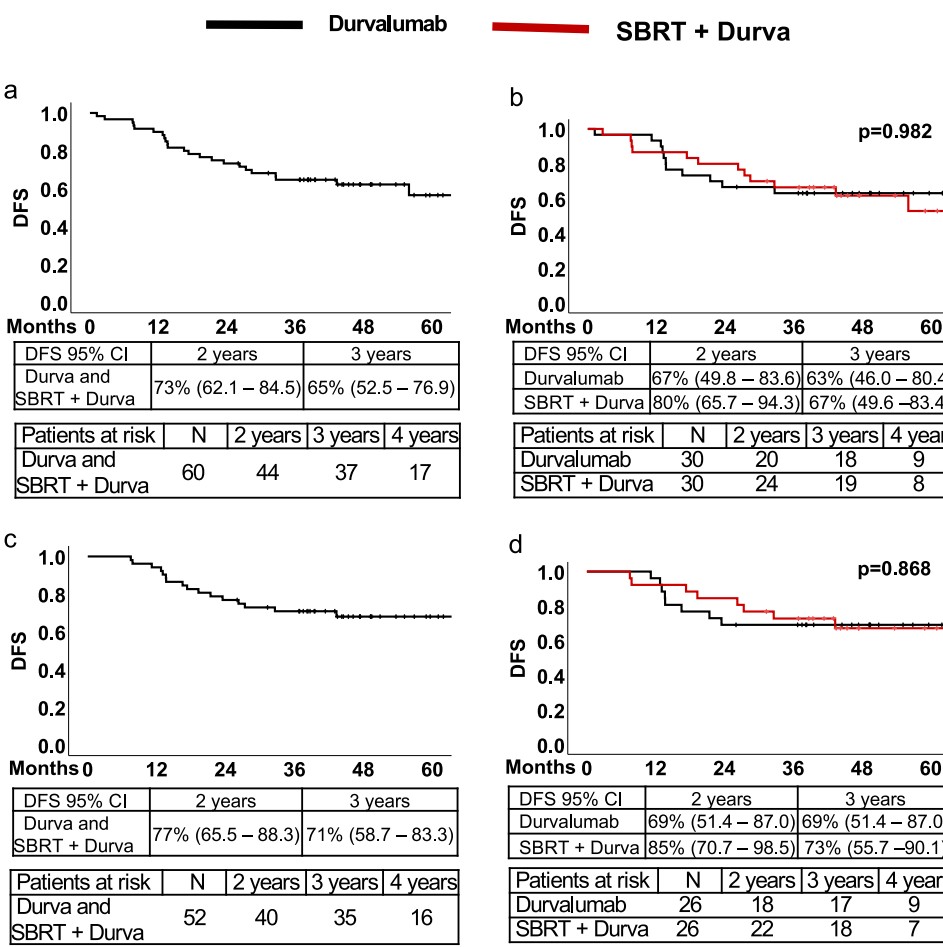

**Fig. 1 | Disease-free survival in all randomized and surgically resected patients.** Disease-free survival for all randomized patients ($n = 60$) (**a**) and in each arm of the trial (**b**). DFS in patients in all surgically resected patients ($n = 52$) (**c**) and each arm (**d**). All panels, Kaplan-Meier survival method. Panels (**b**) and (**d**), log rank test $p$-values. Source data for all panels are provided as a Source Data file.

patients in both arms of the trial, regardless of surgical resection ($n = 60$). After a median follow-up of 49.2 months (range: 44.9–53.6 months), the median DFS for all randomized patients, regardless of treatment assignment, was not reached. Two and three-year DFS were 73% (95% CI: 62.1–84.5) and 65% (95% CI: 52.5–76.9), respectively (Fig. 1a). For the exploratory endpoint of DFS in each arm of the trial, 3-year DFS was 63% (95% CI: 46.0–80.4) in the durvalumab monotherapy arm compared to 67% (95% CI: 49.6–83.4) in the dual therapy arm (Fig. 1b).

In a post hoc exploratory analysis, DFS was estimated in patients who underwent the planned surgical resection. In the surgically resected population ($n = 52$), median DFS was not reached. Two and three-year DFS were 77% (95% CI: 65.5–88.3%) and 71% (95% CI: 58.7–83.3) (Fig. 1c). Three-year DFS was 69% (95% CI: 51.4–87.0) in the durvalumab alone group and 73% (95% CI: 55.7–90.1) in the dual therapy group (Fig. 1d).

**Progression-free survival**

Among the 52 patients who had the planned surgical resection, there were 13 patients who died from either disease progression or other causes of death, 36 who are alive without disease (18 in each arm), and 3 who are alive with recurrence. The causes of death in the surgically resected groups are shown in Table 2. In a post hoc unplanned analysis, we estimated PFS In all 52 surgically resected patients; median PFS was not reached. Two and three-year PFS were 80% (CI: 68.8–91.2) and 76% (CI: 64.2–87.8), respectively (Fig. 2a). Two and three-year PFS were 69%

and 69%, respectively in the durvalumab alone arm and 92% and 83% in the SBRT plus durvalumab arm ($p = 0.190$) (Fig. 2b). Stage-dependent PFS did not differ between arms although survival was numerically though not statistically significantly higher in patients with clinical stage III disease treated by SBRT plus durvalumab (Fig. 2c, d).

In an unplanned post hoc analysis, PFS was compared between patients who achieved MPR and those who did not. Patients who achieved an MPR had a 3-year PFS of 89% (95% CI:74.4–100) compared to a 3-year PFS of 69% (95% CI: 62.7–84.9) in patients who did not achieve an MPR ($p = 0.092$) (Fig. 2e). There were 24 out of 26 resected patients in the monotherapy arm and 10 out of 26 resected patients in the dual therapy arm who did not achieve a major pathological response. Median PFS was not reached in either group, and 3-year DFS appeared similar in both groups (Fig. 2f).

**Table 2 | Causes of deaths in surgically resected patients ($n = 52$)**

| | Monotherapy | SBRT + durvalumab |
|---|---|---|
| Disease | 6 | 3 |
| COVID | 0 | 2 |
| Respiratory failure/pneumonia | 0 | 1 |
| Unknown cause | 0 | 1 |
| Total | 6 | 7 |

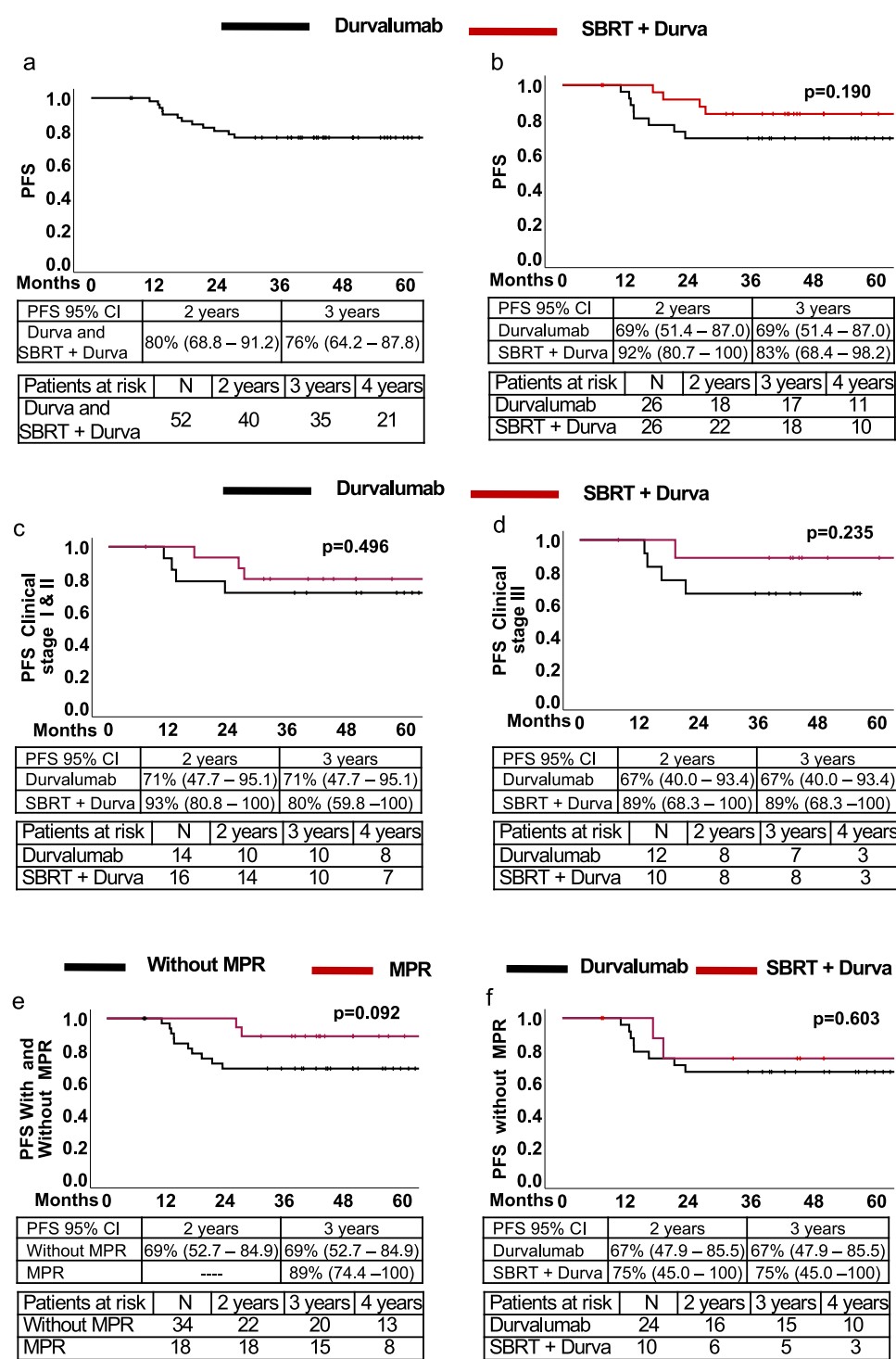

**Fig. 2 | Progression-free survival in all resected patients and stratified by treatment arm, stage and major pathological response.** Progression-free survival for all surgically resected patients (*n* = 52) (**a**) and in each arm of the trial (**b**). PFS in patients with clinical stages I/II (**c**) and clinical stage III (**d**) in each arm. PFS in all 52 patients with and without MPR (**e**). PFS in 34 patients without MPR in each arm of the trial (**f**). All panels, Kaplan-Meier survival method. Panels (**b**) through (**f**), log rank test *p*-values. Source data for all panels are provided as a Source Data file.

## Recurrence patterns

Disease recurrence developed in 12 of 52 resected patients (9 distant, 2 local, and 1 local plus distant). In the durvalumab alone arm, there were 8 of 26 (30.7%) resected patients who developed disease recurrence compared to 4 of 26 (15%) patients who had recurrent cancer in the dual therapy group. Recurrent disease occurred in 2 of 18 patients (11%) with MPR, both of whom had a complete pathological response. In contrast, among 34 patients who did not achieve MPR, 10 patients (29%) developed recurrent lung cancer (monotherapy, 8; dual therapy, 2). Among 34 patients who did not achieve a major pathological response, there was no difference between patients who had recurrent disease and those who did not in baseline demographic characteristics, clinical stage distribution, PD-L1 expression, and percentage of residual viable tumor cells.

**Table 3 | Baseline and clinical variables in patients without MPR who had disease recurrence regardless of treatment arm**

| | Recurrence n = 10 | No recurrence n = 24 |
|---|---|---|
| Age, median (IQI), in years | 68, (63–75) | 73, (65–77) |
| **Gender** | | |
| Male | 4 (40%) | 11 (45.8%) |
| Female | 6 (60%) | 13 (54.2%) |
| **Race** | | |
| White | 7 (70%) | 20 (83.4%) |
| Black | 2 (20%) | 2 (8.3%) |
| Asian | 1 (10%) | 2 (8.3%) |
| **Smoking** | | |
| Never | 5 (50%) | 3 (12.5%) |
| Former/current | 5 (50%) | 21 (87.5%) |
| **Clinical stage** | | |
| IA/IB | 3 (30%) | 10 (41.7%) |
| II/IIIA | 7 (70%) | 14 (58.3%) |
| Clinical tumor size, median (IQI), in mm | 35, (32–71) | 39, (27–51) |
| **Histology** | | |
| Adenocarcinoma | 8 (80%) | 14 (58.3%) |
| Squamous | 2 (20%) | 7 (29.2%) |
| Other | 0 | 3 (12.5%) |
| **Percent PDL1** | | |
| negative | 6 (60%) | 12 (50%) |
| ≥1% and <25% | 3 (30%) | 6 (25%) |
| ≥25% | 1 (10%) | 4 (16.7%) |
| Unknown | 0 | 2 (8.3%) |
| **EGFR** | | |
| Positive | 5 (50%) | 4 (16.7%) |
| Negative | 5 (50%) | 20 (83.3%) |
| **Percent residual tumor** | | |
| ≤25 | 2 (20%) | 7 (29.2%) |
| >25 to ≤50 | 5 (50%) | 8 (33.3%) |
| >50 to ≤75 | 2 (20%) | 7 (29.2%) |
| >75 to ≤90 | 1 (10%) | 2 (8.3%) |
| **Adjuvant chemotherapy ± radiation** | | |
| Chemo and RT | 5 (50%) | 6 (25%) |
| Chemo alone | 3 (30%) | 8 (33.3%) |
| None | 2 (20%) | 10 (41.7%) |
| **Received adjuvant durvalumab** | | |
| No | 5 (50%) | 10 (41.7%) |
| Yes | 5 (50%) | 14 (58.3%) |

However, patients who had disease recurrence were more likely to have had EGFR-mutated tumors and more likely to have had adjuvant chemotherapy (Table 3).

## Correlation of pretreatment peripheral blood CD103+ T cells with MPR

In a post hoc unplanned analysis, we explored the association between peripheral blood immunophenotype and clinical outcome. Peripheral blood mononuclear cells (PBMCs) were obtained from study participants at defined time points: pretreatment, immediately pre-operatively, and 3 months postoperatively. Using a 23-marker spectral flow analysis and established dimensionality reduction and clustering tools, we identified T cells with hallmarks of prior antigen exposure (CCR7−, CD45RA−) and tissue residency (CD103+)[25] in both the CD4 (cluster CD4

P1) and CD8 (clusters CD8 P7 and P8) compartments, suggesting tumor-experienced T cells were recirculating into the blood (Fig. 3A, F). Of the two CD8 clusters, we focused our analysis on CD8 P7, given its preserved expression of CD28, CD27, and TCF1, which suggests preserved effector function and capacity for self-renewal (Fig. 3F)[26–28]. The correlation between pretreatment CD103+ T cells (CD4 P1, CD8 P7, and P8) and the presence or absence of MPR defined as ≥90% tumor regression is shown in Fig. 3 where data is presented stratified by treatment arm as well as for all patients regardless of treatment assignment. Patients with MPR after neoadjuvant therapy had significantly greater pretreatment frequencies of these CD103-expressing populations (Fig. 3B, C, G, H, K, L). This finding was most prominent in patients in the dual therapy arm, where there was a higher number of patients achieving MPR (Fig. 3B, C, G, H, K, L). Using manual gating on the samples, we found that gating on CD103+ CD4 and CD8 T cells produced similar significant results, suggesting that CD103 expression is the primary defining marker of this cell population (Fig. 3D, E, I, J). Notably, elevated pretreatment frequency of PD1+ T cells was also significantly associated with MPR (Fig. 3M, N). We also found similar trends present for these populations at the 3-month postoperative timepoint (Supplementary Fig. S2). Together, these data show the presence of an unexpected population of T cells in the peripheral blood with features of tissue residency, whose frequency is correlated with MPR and, thus, may serve as a predictive biomarker in the context of neoadjuvant treatment of NSCLC.

## Correlation of postoperative PD1+ T cells and CD103+ CD4 T cells with disease recurrence

We next stratified study participants into two groups based on their most recent disease status—those with no evidence of disease and those with disease progression. We found that the frequency of PD1+ T cells, as well as the frequency of CD103+ CD4 and CD8 T cells, at 3 months postoperatively was greater in patients without progression irrespective of treatment arm (Fig. 4A–D). Given the difference in recurrence rates between patients with and without MPR, we also investigated whether these T cell populations were predictive of recurrence in patients who did not have an MPR (Supplementary Fig. S3). We found a nonsignificant trend toward greater frequency of PD1+ T cells at 3 months post-resection in patients who did not have an MPR but remained disease-free at the conclusion of the trial (Supplementary Fig. S3).

To assess the association of PD1+ T cells with recurrence as suggested by flow cytometry, we performed multiplex immunofluorescence analyses on seven resected tumors from patients who did not achieve MPR and who remained disease-free and five similar patients who subsequently developed recurrence of disease. There were no differences between the two groups in tumor cell density, tumor cell ki67 expression, or percent of tumor cells expressing PD-L1 (Supplementary Fig. S4A–C). Although there was no difference in the densities of CD3+ cells between the groups, there was a trend toward increases in CD8+PD1+, CD4+PD1+ cells, and both together in tumors that did not recur (Supplementary Fig. S4D–F). These findings are consistent with the differences observed in the profiling of T cells in peripheral blood (Fig. 4).

In aggregate, the data suggest that a higher frequency of PD1+ T cells in the peripheral blood at 3 months postoperatively may be correlated with freedom from recurrence and thus warrant validation in larger trials.

## High levels of post-treatment intra-tumoral CD103+ T cells are correlated with freedom from recurrence regardless of MPR

We also conducted a post hoc analysis to determine the correlation between gene expression profiles in tumors and clinical outcomes. Based on the above findings, we interrogated bulk RNAseq gene expression data from pretreatment biopsy and resected tumors for

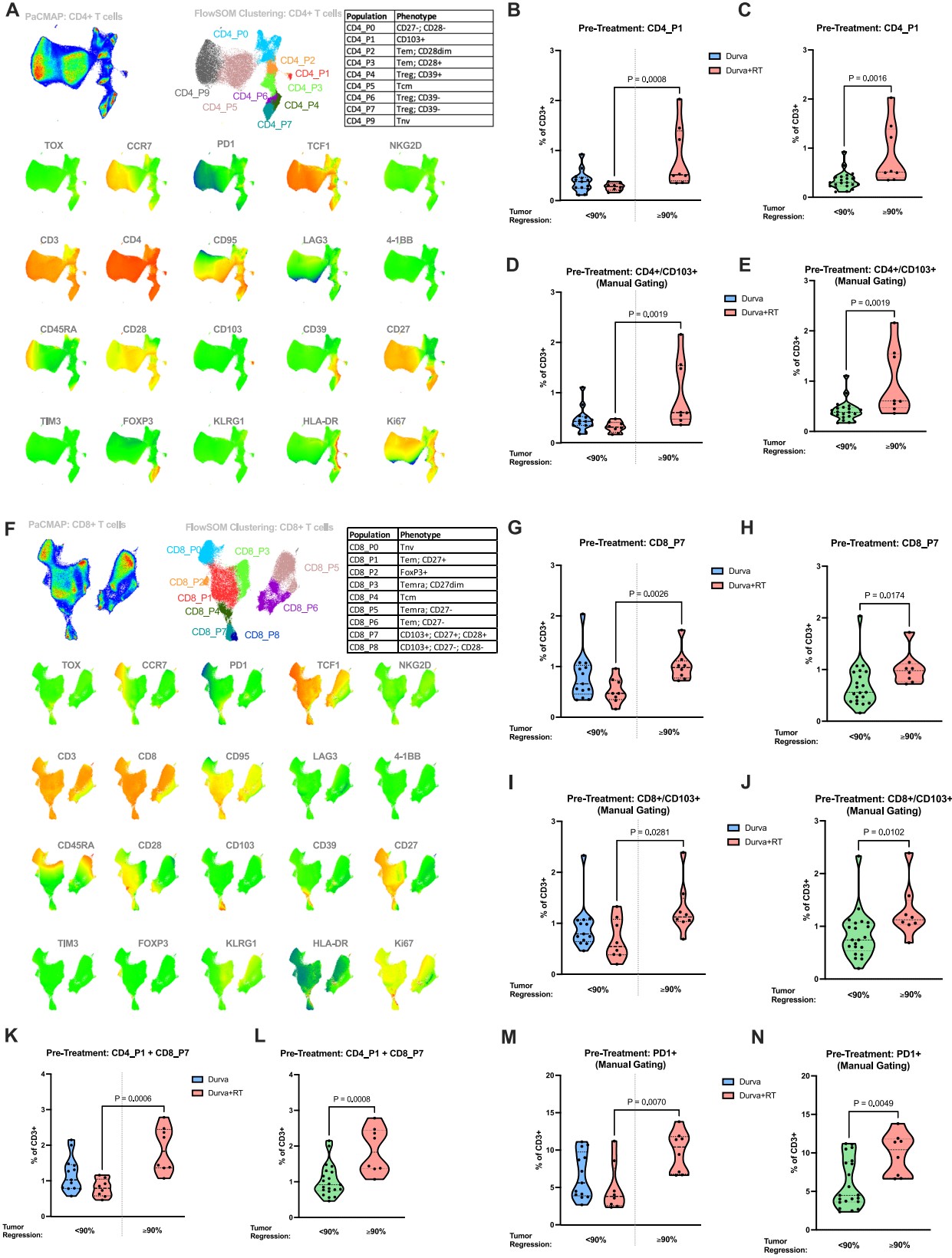

differences in CD103 gene (*ITGAE*) expression. There was no post-treatment (at resection) change in CD103 gene expression in the durvalumab alone arm (Fig. 5a). There was a post-treatment increase in CD103 gene expression in the dual therapy arm only for tumors that achieved MPR (Fig. 5b, c). These findings are consistent with a published report that an increase in intra-tumoral CD103 gene expression

is associated with response to PD-L1 blockade[29]. To further assess the potential correlation between intra-tumoral CD103 expression and subsequent disease recurrence, we compared CD103 expression from resected tumor samples obtained from patients who either remained disease-free ($n = 20$) or subsequently developed disease recurrence ($n = 9$). Intra-tumoral CD103 gene expression was significantly higher

**Fig. 3 | Association of pretreatment circulating T cell populations with MPR (≤90% residual viable tumor cells). A** PaCMAP projection and FlowSOM clustering of CD4+ T cells from all PBMC samples. Heatmap overlays indicates relative expression for each phenotypic marker. Tnv = naïve T cell phenotype, $T_{reg}$ = regulatory T cell phenotype, $T_{em}$ = effector memory T cell phenotype, $T_{cm}$ = central memory T cell phenotype. In all panels, patients are stratified by percent tumor regression: those with ≥90% tumor regression (defined clinically as major pathological response) and those with <90% tumor regression. Comparison between treatment arms shown in panels (**B, D, G, I, K, M**). Comparison between MPR and no MPR regardless of treatment arm shown in panels (**C, E, H, J, L, N**). **B, C** Frequency of CD4_P1 (CD103-expressing) T cell population in PBMC samples. **D, E** Frequency of CD103+/CD4+ T cells in PBMC samples measured by manual gating. **F** PaCMAP projection and FlowSOM clustering of CD8+ T cells from all PBMC samples. Heatmap overlays indicates relative expression for each phenotypic marker. Tnv = naïve T cell phenotype, $T_{emra}$ = terminal effector T cell phenotype, $T_{em}$ = effectory memory T cell phenotype, $T_{cm}$ = central memory T cell phenotype. **G, H** Frequency of CD8_P7 (CD103-expressing) T cell population in PBMC samples. **I, J** Frequency of CD103+/CD8+ T cells in PBMC samples measured by manual gating. **K, L** Combined per-patient frequency of CD8_P7 and CD4_P1 T cell populations in PBMC samples. **M, N** Frequency of PD1+ T cells in PBMC samples measured by manual gating. All panels, two-tailed Mann-Whitney test. Panels (**B, D, G, I, K, M**) Durva <90% ($n = 13$); Durva + SBRT < 90% ($n = 8$); Durva ≥90% ($n = 0$); Durva + SBRT, ≥90% ($n = 8$). Panels (**C, E, H, J, L, N**): <90% ($n = 21$); ≥90% ($n = 8$). Source data for all panels are provided as a Source Data file.

in tumor samples from patients who remained disease-free compared to tumors from patients who had disease recurrence, independent of the presence of a major pathological response (Fig. 5d). These data suggest that high levels of intra-tumoral CD103+ T cells, even in the absence of MPR, may be an early correlate of freedom from recurrence.

### In the absence of MPR, upregulation of immune-related genes is associated with freedom from recurrence

The increase of intra-tumoral CD103 expression in resected tumors of patients without MPR who did not develop cancer recurrence prompted us to interrogate the RNAseq data for other differences in gene expression associated with recurrence-free survival among those patients. We analyzed the post-treatment bulk tumor gene expression profiles of 29 of 31 patients who did not achieve MPR and for whom we had post-treatment RNAseq data: 9 patients who developed disease recurrence and 20 patients who did not. There were no differences in the percent residual viable tumor, percent of PD-L1+ cancer cells, or tumor mutational burden between the two groups (Fig. 6a–c). This was also the case when tumors were grouped by treatment arm (Supplementary Fig. S5). Analyses of bulk RNAseq revealed no individual gene differentially expressed between tumors from individuals free of recurrence and those that had cancer recurrence (false discovery rate (FDR) < 0.1, Benjamini-Hochberg correction). However, to explore the bulk RNAseq data for differences in biological pathways between the groups, we explored the 300 genes with the lowest raw $p$-values (not corrected for multiple testing), using Gene Ontology pathway enrichment analysis (Fig. 6 and Supplementary Data 1). Of the 170 genes that increased in expression in tumors from patients without recurrence, nearly a third of the genes (29%) are annotated to the Gene Ontology immune system process (GO:0002376), with enrichments of different aspects of immune regulation, immune cell migration, and extracellular matrix biology (Fig. 6e). There were no enriched Gene Ontology pathways among the 130 genes with non-adjusted $p$-values whose expressions were increased in patients with cancer recurrence. Comparison of the tumor immune microenvironments between tumors of patients with and without cancer recurrence, by deconvolution of the bulk RNAseq, revealed no significant differences between the groups in immune cell populations identified by xCell deconvolution.

### Discussion

We had previously reported that compared to neoadjuvant durvalumab alone, neoadjuvant durvalumab combined with focal sub-ablative doses of stereotactic radiation is safe and associated with a significant and clinically meaningful increase in the proportion of patients achieving a major pathological response independent of tumoral PD-L1 expression[24]. Here we report on the prespecified secondary endpoint of disease-free survival and (in a post hoc unplanned analysis) on progression-free survival after a median follow-up of 49 months. Two and 3-year DFS for all 60 randomized patients were 73% and 65%, respectively. In the surgically resected population, the corresponding 2 and 3-year DFS were 77% and 71%, respectively. These survival

outcomes appear promising, particularly since 25% and 43% of patients presented with stage II and III disease, respectively. Survival outcomes also appeared favorable when the analysis was restricted to surgically resected patients who developed lung cancer-related events (recurrence or death from cancer). In all surgically resected patients, 2 and 3-year PFS were 80% and 76% respectively. Treatment with neoadjuvant SBRT and durvalumab was associated with an improved PFS compared to patients treated by neoadjuvant durvalumab alone (3-year PFS: 83% vs 69%, p = 0.190). Although the intergroup difference in PFS did not cross the prespecified statistical boundary, it is encouraging that the PFS curves separated at 12 months and maintained a 20% relative difference in survival at 24 months and beyond. The difference in PFS was more apparent in patients with stage III disease; however, the small sample size in each stage cohort precludes any meaningful conclusions. Predictably, patients who achieved a major pathological response had a strong trend toward an improved PFS compared to those who did not. PFS at 3 years was 89% in patients who had an MPR compared with 69% in patients who did not achieve MPR.

Interestingly, recurrence of the index cancer occurred in 11% of patients who achieved MPR, emphasizing the importance of continued surveillance of these patients for evidence of disease progression. It is equally interesting that patients who did not achieve MPR had an unexpectedly favorable 3-year PFS of 69%, particularly since over 60% had stage II or III disease. These findings suggest that MPR (including complete pathological response) may not be the sole driver of improved survival after neoadjuvant immunotherapy. For example, it is possible that even in the absence of an immune response sufficiently robust to eradicate the primary tumor, immunotherapy may induce a systemic adaptive immune response that stabilizes or eliminates micro-metastases. In support of this hypothesis, our data identify signals of an enhanced systemic immune response associated with freedom from disease recurrence in patients in whom neoadjuvant therapy did not result in a major pathological response. In these patients, neoadjuvant immunotherapy was associated with a significant increase in CD103+ T cells in resected tumors as well as intratumoral increased expression of immune-related genes following treatment.

Interestingly, there was a nonsignificant trend toward a higher frequency of circulating PD1+ CD4+ CD8+ T cells detected 3 months postoperatively in patients without subsequent recurrence, even in the absence of MPR. The systemic immune response linked to an enhanced local anti-tumor immune response is reminiscent of in situ vaccination; a systemic anti-tumor response arising from immunologic cell death of cancer cells and or stimulation of immune response within the tumor[18,23]. Although SBRT may induce in situ vaccination, resulting in systemic, long-lasting anti-tumor immunity[23], we found that an intra-tumoral and systemic immune response in the absence of MPR was not restricted to the SBRT/durvalumab therapy arm.

Our analysis of pretreatment peripheral blood T cell phenotype revealed the presence of a circulating population of CD4+ and CD8+ T cells that express CD103, a marker of tissue-resident tumor-infiltrating lymphocytes that are rarely found in peripheral blood and that

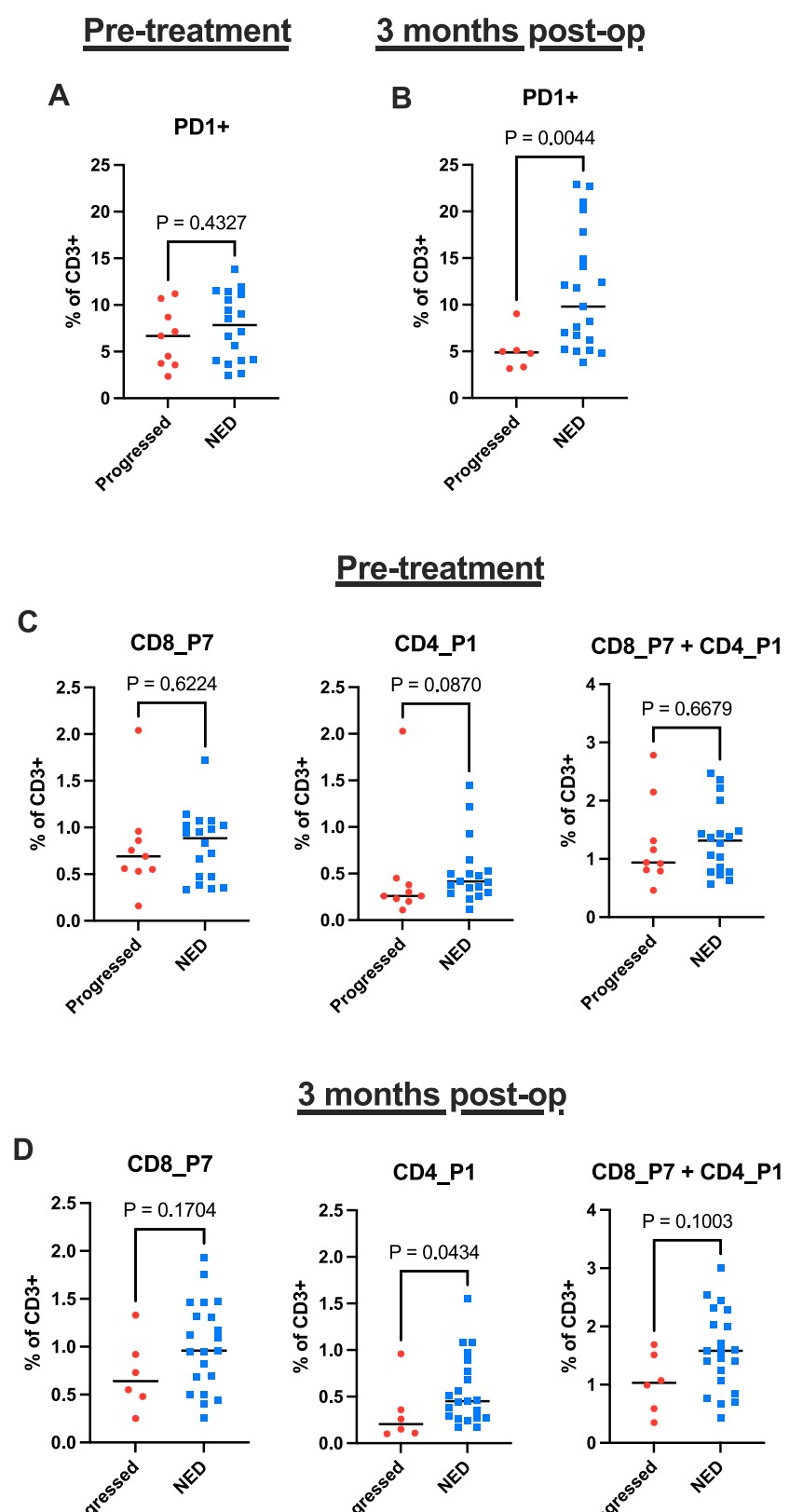

**Fig. 4 | Association of circulating T cell populations with long-term disease-free survival.** Study participants were stratified into two groups based on their disease status at the conclusion of the trial: those with no evidence of disease (NED) and those with progression (died from disease or alive with disease). These groups were compared for frequency of circulating: **A**, **B** PD1⁺ T cells, **C**, **D** CD8_P7 and CD4_P1

(CD103-expressing) T cell populations. **A–D** Two-tailed Mann-Whitney test. **A** Progressed ($n=9$), NED ($n=18$); **B** Progressed ($n=5$), NED ($n=21$); **C** all panels, Progressed ($n=9$), NED ($n=18$). Source data for all panels are provided as a Source Data file.

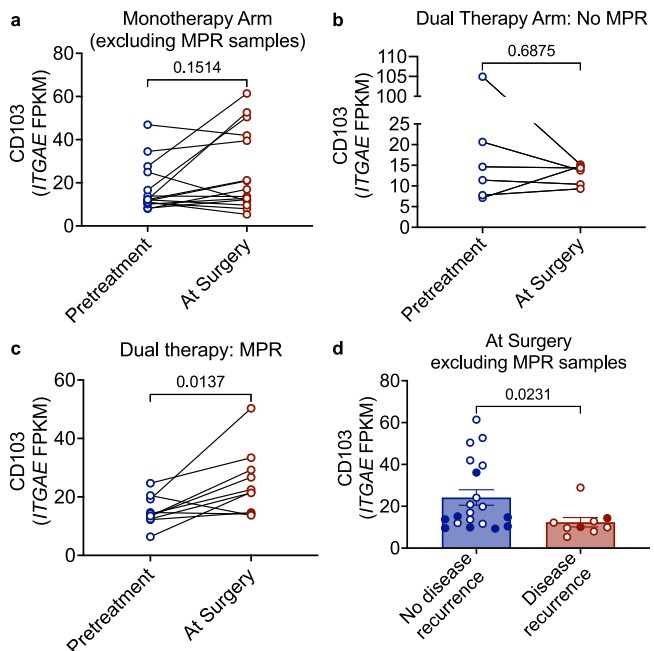

**Fig. 5 | CD103 gene (*ITGAE*) expression. a** CD103 (*ITGAE*) gene expression (FPKM) from bulk RNAseq of pretreatment (tumor biopsy) and at surgery (resected tumor) for monotherapy cases by MPR (*n* = 15 matched samples). Two-sided, paired Mann-Whitney test. **b** CD103 (*ITGAE*) gene expression (FPKM) from bulk RNAseq of pretreatment (tumor biopsy) and at surgery (resected tumor) for dual therapy cases that did not achieve MPR. Two-sided, paired Mann-Whitney test (*n* = 6 matched samples). **c** CD103 (*ITGAE*) gene expression (FPKM) from bulk RNAseq of pretreatment (tumor biopsy) and at surgery (resected tumor) for dual therapy cases that achieved MPR. Two-sided, paired Mann-Whitney test (*n* = 10 matches samples). **d** CD103 (*ITGAE*) gene expression (FPKM) from bulk RNAseq of resected tumors (at surgery), excluding MPR samples, with (*n* = 9) and without (*n* = 20) disease recurrence. Solid symbols are from the dual therapy arm. Mean ± SEM. Two-sided, unpaired Mann-Whitney test. Source data for all panels are provided as a Source Data file.

are enriched for tumor-reactive T cells with a distinct T-cell receptor (TCR) repertoire[30]. CD103 (encoded by *ITGAE*) is a classical marker of tissue-resident memory T cells (T$_{RM}$) and functions as an adhesion molecule binding to E-cadherin[31]. T$_{RM}$ reside in large barrier tissues (e.g., skin, lung, gut) and confer organ-wide immune responses following a local immune stimulus[32,33]. An increase in the proportion of intra-tumoral CD103+ and T$_{RM}$-like tumor-infiltrating lymphocytes is associated with improved responses to immune checkpoint inhibition as well as improved survival in a variety of tumors, including melanoma, lung cancer, head and neck, and bladder cancer[25,29,30,34]. Because CD103 typically indicates tissue-residing T cells, identifying these cells in the circulation is surprising. Recent studies demonstrate that partly differentiated T cells circulate from lymph nodes to tumors, where they acquire tissue residency and effector differentiation to promote anti-tumor immunity in response to immune checkpoint blockade[35]. Thus, it is possible that the CD103+ T cells captured in blood in our study may represent this pool of recently (re-)circulating T cells, and a higher rate of this process may represent a substrate for improved anti-tumor immunity. Indeed, a recent report by Nose and colleagues reported that in patients with advanced gastric cancer, those with a high frequency of CD8+ PD-1+CD103+ T cells in peripheral blood two weeks after the start of treatment had significantly better response to immune checkpoint inhibition and better progression-free survival[36]. We found that a higher frequency of circulating CD103+ expressing CD4+ and CD8+ T cells in pretreatment samples is correlated with the subsequent development of major pathological response, suggesting that high pretreatment frequency of CD103+ T cells may be a useful

predictive biomarker of response to immunotherapy. The association of high pretreatment frequency of CD103+CD8+ and CD4+ T cells with major pathological response to therapy suggests that responding patients are likely to have an antecedent, nascent, albeit ineffective, anti-tumor immune response. Further characterization of this population in larger cohorts is highly feasible, requiring simple blood draws, and may serve as a biomarker to guide therapy.

Furthermore, in patients without MPR, approximately 70% did not develop cancer recurrence. In these patients, we find a nonsignificant trend toward greater PD1+ T cells at 3 months post-resection. This suggests that these cells may contribute to the control of the primary tumors as well as distant microscopic diseases and warrant validation in larger trials. The association of a peripheral T cell response with freedom of recurrence is also supported by the results of gene expression profiles in patients who did or did not develop disease recurrence. We found that in the absence of MPR, genes annotated to the immune system were significantly upregulated in patients who remained disease-free. Intra-tumoral CD103 (*ITGAE*) gene expression, as well as the increase in the numbers of CD8+CD4+ PD1+ T cells, further support these findings. The transcriptional changes supporting an enhanced immune response associated with freedom from recurrence are based on a hypothesis-generating analysis of the data and, as such, need to be confirmed in future studies. Nonetheless, the gene expression profiling is consistent with an active immune response including upregulation of markers of immune activation such as TIM-3 (*HAVCR2*) and *CD70* (expressed on antigen-activated immune cells), *STK10* (a marker of tumor-infiltrating immune cells) as well as genes associated with antigen-presenting cells (*CD68, FPR1, FPR2, FCGR2A, TMEM176A, TMEM176b*).

In aggregate, our findings support several important conclusions. First, our neoadjuvant strategy of low-dose SBRT plus immune checkpoint blockade was associated with favorable DFS and PFS, which seem comparable to those observed after chemo-immunotherapy approaches. We also observed that MPR, including complete pathological response, though important, may not be the only determinant of improved survival after neoadjuvant immunotherapy. Second, the presence of circulating antigen-experienced CD103+ T cells before treatment in patients who achieve an MPR after neoadjuvant therapy suggests their potential utility as predictive biomarkers. Additionally, the presence of pretreatment CD103+ T cells also suggests that these patients may have a pre-existing suboptimal immune response that is further potentiated by neoadjuvant therapy. Finally, we found that in patients not achieving MPR, assessment of immune-related gene expression in resected tumors, levels of intra-tumoral CD103+ T cells, as well the frequency of circulating PD1+ T cells 3 months postoperatively, may provide an early signal of the probability of subsequent disease recurrence thus allowing de-escalation (or modification) of therapy before the emergence of clinically evident recurrent disease.

The interpretation of our findings is limited by the small sample size and the use of an admittedly uncommon regimen of neoadjuvant therapy. Furthermore, the trial was appropriately powered for the primary endpoint of MPR but not for any of the secondary or exploratory endpoints. Our survival results are also limited by and likely negatively impacted by the relatively high proportion of patients with EGRF-mutated tumors, though those appear to be reasonably distributed between arms. Therefore, our clinical and correlative results must be construed as hypothesis-generating and thus require testing and validation in larger controlled appropriately powered trials.

Novel neoadjuvant immunotherapy strategies using immune checkpoint blockade are currently being evaluated in ongoing trials. For example, the NeoCOAST-2 trial (NCT05061550), a platform neoadjuvant trial, will evaluate the efficacy of multiple neoadjuvant immune checkpoint inhibitors such as the anti-CD73 monoclonal antibody oleclumab, the anti-NKG2A antibody monalizumab; and the PD-1/CTLA-4 bispecific antibody volrustomig in combination with

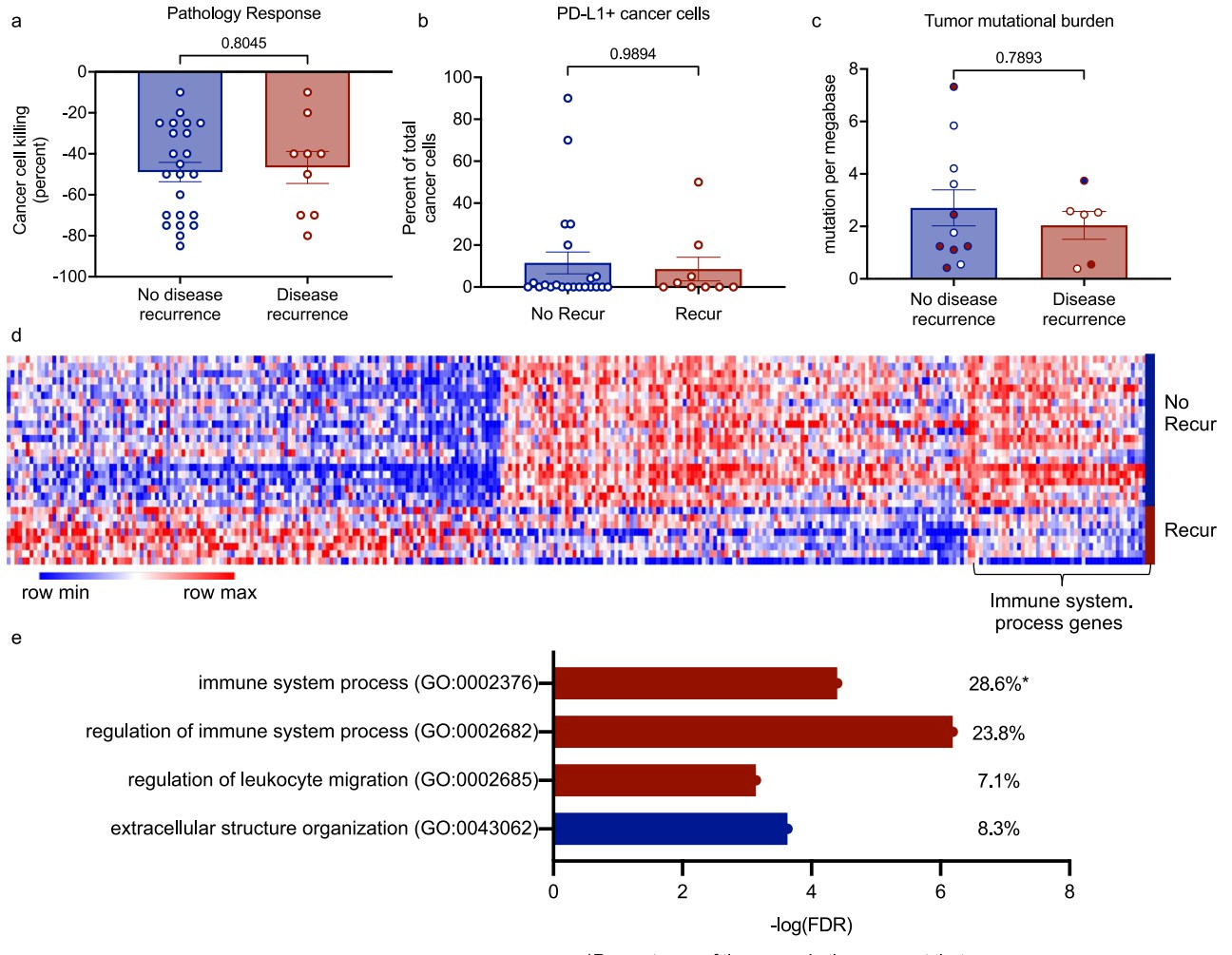

**Fig. 6 | Characteristics of tumors that do not recur. a** Pathology response, as percent of cancer cell killing, for tumors without MPR grouped by disease recurrence. Mean ± SEM. No recur ($n = 23$) and Recur ($n = 9$). Two-sided, unpaired Mann-Whitney test. **b** PD-L1 positive cancer cells as percent of total cancer cells determined by IHC for tumors without MPR grouped by disease recurrence. Mean ± SEM. No recur ($n = 22$) and recur ($n = 9$) Two-sided, unpaired Mann-Whitney test. **c** Tumor mutational burden for tumors without MPR grouped by disease recurrence. Mean ± SEM. No recur ($n = 11$) and recur ($n = 6$). Two-sided, unpaired Mann-Whitney test. **d** Heatmap of differentially expressed genes between tumors with or without disease recurrence, excluding tumors that achieved MPR. Solid symbols are from the dual therapy arm. **e** Some GO pathways enriched among genes upregulated in tumors that did not recur. Percentages of the genes in the gene set that were differentially expressed in the data. Source data for panels (**a**, **b**, **c**, **e**) are provided as a Source Data file.

chemotherapy in patients with resectable, early-stage NSCLC. In contrast, a chemotherapy-free regimen is being evaluated by SKYSCRAPER-05 (NCT03563716), a global neoadjuvant phase II trial of the anti-PD-L1 antibody atezolizumab plus the anti-TIGIT antibody tiragolumab with or without platinum-based chemotherapy in patients with locally advanced resectable stage II–IIIB NSCLC. Given the current use of neoadjuvant chemotherapy in combination with immune checkpoint inhibitors, our trial results raise the question of which strategy may be most optimal. We are currently in the late planning stages of a randomized trial comparing our current regimen of SBRT plus PD1/PD-L1 blockade with chemo-immunotherapy, the current standard of care. This will also provide an important and necessary opportunity to prospectively test the role of the discovered blood- and tissue-based biomarkers shown here.

## Methods
### Trial design and participants
We performed a single center, investigator-initiated randomized phase II neoadjuvant trial of the anti-PD-L1 antibody durvalumab alone (Arm I) or in combination with SBRT (Arm II) in patients with NSCLC clinical stages I-IIIA NSCLC according to the seventh edition of the American Joint Committee on Cancer TNM staging system. The trial protocol was approved by the Institutional Review Board of Weill Cornell Medicine and the New York Presbyterian Hospital (protocol number 15010157950), and the trial was monitored by the Weill Cornell Medicine Data Safety Monitoring Board. The trial was done in accordance with the International Conference on Harmonization Guidelines on Good Clinical Practice and the Declaration of Helsinki. This trial was registered with ClinicalTrial.gov on September 19, 2016, and is ongoing but closed to accrual (https://classic.clinicaltrials.gov/ct2/show/NCT02904954).

Patients were enrolled regardless of smoking history, PD-L1 expression or results of tumor genotyping. Patients who had any of the following were considered ineligible for the trial: concurrent invasive malignancy, history of another invasive cancer within the past 3 years, active autoimmune disease, systemic immune suppression, and radiographic evidence of interstitial lung disease. All patients provided written informed consent for participation in the clinical trial

as well as for utilization of their biospecimens in biomarker studies. Patients could withdraw consent at any time for any reason.

## Randomization and masking

Permuted blocked randomization with varied block sizes was generated for the trial in a 1:1 allocation ratio with no stratification for clinical or molecular variables. Block sizes were concealed from all investigators/study personnel and were only known by the study statistician (PC). The trial assignment was unblinded/unmasked by design; patients, treating physicians, and all study personnel were aware of what trial arm the patient was enrolled in after the patient was assigned via the blocked allocation sequence.

## Procedures

The first patient was enrolled on January 13, 2017, and the last patient on July 28, 2020. All patients underwent complete clinical staging using computerized tomography (CT) and positron emission tomography (PET) scanning, as well as brain imaging using magnetic resonance. Whenever possible, samples from pretreatment biopsies and post-surgical material were stored for later determination of PD-L1 expression using immunohistochemistry (SP 263, Ventana Medical Systems Inc. Arizona), tumor mutational burden and bulk gene expression profiling using RNA sequencing. Blood and plasma, for determination of circulating immune profile, were collected at baseline, prior to surgical resection as well as 3–6 months postoperatively.

All patients received two cycles of durvalumab 3 weeks apart at a dose of 1.12 g by intravenous infusion over 60 min. Patients in the radiation and durvalumab arm were also treated with SBRT delivered using 3 consecutive daily fractions of 8 Gy initiated immediately preceding the first cycle of durvalumab (same day). The selected radiation dose of 24 Gy is equivalent to a biologically effective dose (BED) of 43.2 Gy, a significantly lower dose than the standard ablative dose for T1-T2 NSCLC (BED > 100 Gy) and is considered insufficient for tumor ablation[37,38]. Details of radiation planning and delivery, as well as adverse events associated with neoadjuvant therapy, were previously reported[21]. Following preoperative treatment, all patients were restaged 1–2 weeks after the second cycle of durvalumab.

In the absence of systemic disease progression, surgical exploration was performed within 2–6 weeks following the second cycle of durvalumab. Surgical resection included a lobectomy, bi-lobectomy or pneumonectomy, along with a complete mediastinal node dissection. Postoperatively, all the patients were offered conventional adjuvant chemotherapy as clinically indicated and offered the option of receiving postoperative durvalumab monthly for 12 cycles. Patients were assessed for disease recurrence using CT scanning every 6 months for 2 years and then yearly thereafter.

## Outcome

The primary endpoint, MPR proportion between both arms of the trial, was previously reported[24]. The secondary endpoint was 2-year disease-free survival for the whole cohort. Disease-free survival was defined as the time from randomization to recurrence of lung cancer and/or death from any cause, whichever occurred first. To specifically determine the impact of treatment on oncologic outcomes, we conducted an unspecified post hoc analysis assessing progression-free survival (PFS) in all patients who underwent the planned surgical resection. Progression-free survival was defined as the interval from surgical resection to lung cancer recurrence or death from lung cancer. In addition to assessing PFS for all 52 resected patients, PFS was compared between patients in both arms of the trial, between arms in patients with clinical stages I/II and patients with clinical stage III disease and finally between patients with and without MPR. Unplanned exploratory analyses included a comparison of recurrence rates and patterns of recurrence between arms. In unplanned post hoc analyses, we performed correlative studies correlating MPR and

recurrence with gene expression profiles and peripheral blood immune profiles.

## Correlative studies

For gene expression profiling, libraries for sequencing were prepared from FFPE-isolated RNA using the NEB/Twist RNA Capture library method (New England Biolabs, Ipswich, MA). The libraries were sequenced with PE 2x51bp sequencing, 8 samples per lane (>50 M mapped reads) on a HiSeq4000. Bulk RNAseq was analyzed using standard tools, including analysis of QC metrics, expression quantification using STAR/HTSeq/Cufflinks, differential expression using DESEq2 and deconvolution using xCell[39].

For circulating immune phenotyping, viable PBMCs were isolated at each timepoint and stored for profiling using 23-marker spectral flow analysis of the T cell and myeloid compartments.

## Multiplex imaging

Immunofluorescence imaging of a subset of resection samples (eleven tumors) was performed using the Neogenomics platform as previously described[40,41]. Briefly, formalin-fixed tissue slides were baked at 65 °C for 1 h. Slides were deparaffinized with xylene, rehydrated by decreasing ethanol concentration washes, and then processed for antigen retrieval. A two-step antigen retrieval was adopted to allow antibodies with different antigen retrieval conditions to be used together on the same samples[42]. Samples were blocked against non-specific binding with 10% (wt/vol) donkey serum and 3% (wt/vol) bovine serum albumin (BSA) in phosphate-buffered solution (PBS) for 1 h at room temperature and stained with DAPI for 15 min. Directly conjugated primary antibodies (Cy3 or Cy5) were diluted in PBS supplied with 3% (wt/vol) BSA to optimized concentrations and applied for 1 h at room temperature on a Leica Bond III Stainer. Following derivatization, optimized staining conditions were empirically determined for each batch of each batch of labeled antibody. Antibodies: CD3 (F7.2.38), Dako #M7524; CD4 (EPR6855), Abcam ab181724; CD8 (C8/144B), Dako #M7103; Ki67 (SP6), Abcam #ab16667; PD-1 (EPR4877(2)), Abcam ab186928; PD-L1 (SP142), ab236238, panCK (PCK-26) Sigma-Aldrich #C5992.

Stained images were collected on an INCell analyzer 2200 microscope (GE Healthcare Life Sciences) equipped with high-efficiency fluorochrome-specific filter sets for DAPI, cy3 and cy5[40]. For multiplexed staining where co-localization was desired, the regions of interest (-0.4–0.6 mm² tissue area) were imaged, and stage coordinates were saved. The coordinates of each image region were then recalled for each subsequent round after minor readjustment using reference points from the first-round DAPI image and determining the appropriate offset. The exposure times were set at a fixed value for all images of a given marker. For image analyses, microscopy images were exported as full-resolution TIFF images in grayscale for each individual channel collected.

## MultiOmyx image analytics

The acquired images from sequential rounds were registered using DAPI images acquired in the first round of staining via a rigid registration algorithm for each region of interest. The parameters of transformation were then applied to the subsequent rounds, which ensured that the pixel coordinates across all the imaging rounds corresponded to the same physical locations on the tissue. Classification and co-expression analysis were performed in multiple stages. First, a nuclear segmentation algorithm was applied to the DAPI image to delineate and identify individual cells. Location information and expression of all the markers were computed for every cell identified. Then, morphologic image analysis and shape detection were performed using proprietary algorithms (Neogenomics Laboratories; https://neogenomics.com/pharma-services/lab-services/multiomyx). These algorithms detect and classify cells as positive or negative for

each marker depending on their subcellular localization and morphology. A tissue-quality algorithm was also applied to the images to ensure image artifacts that arose owing to tissue folding or tear did not affect cell classification. Co-expression analysis and phenotype identification were performed by combining individual marker classification results.

## Culture of PBMC samples

Frozen PBMCs from patients treated by neoadjuvant durvalumab alone were available from 13 patients pretreatment, 17 patients prior to surgery and 13 patients at 3 months postoperatively. All were from patients who did not achieve MPR except for two of the 3 months postoperative samples. In patients treated by SBRT plus durvalumab, frozen PBMCs were available from 16 patients pretreatment (8 that did not achieve MPR and 8 that did), 16 patients at surgery (7 that did not achieve MPR and 9 that did) and 14 patients 3 months post-therapy for (4 that did not achieve MPR and 10 that did). PBMC samples were each thawed rapidly in a water bath at 37 °C, resuspended in 10 mL RPMI 1640 medium (Gibco), and centrifuged for 5 min at 400×$g$. Samples were then washed 1x with PBS and resuspended in RPMI 1640 supplemented with 10% fetal bovine serum (Gibco), 2 mM GlutaMAX (Gibco), 1% Penicillin/Streptomycin (Gibco), and 200 IU/mL rhIL2 (Chiron). Samples were then plated in 24-well plates and incubated overnight at 37 °C with 5% $CO_2$. The following day, all PBMC samples were collected for staining and flow cytometry analysis.

## Flow cytometry

Following overnight culture, all PBMC samples were collected in 5 mL Eppendorf tubes. Samples were centrifuged at 400×$g$ for 5 min, washed once with ice-cold PBS, and transferred to 96-well plates for staining. All samples were resuspended in 100 μL 1:500 Zombie-NearIR (Biolegend) in PBS and stained for 15 min on ice in the dark. Samples were then washed once with ice-cold FACS buffer (PBS with 2% FBS and 2 mM EDTA; Gibco), blocked for 10 min with 1:20 Human TruStain FcX (Biolegend) in FACS buffer, and then stained with surface antibodies in FACS buffer supplemented with 10 μL Brilliant Stain Buffer Plus (BD) (100 μL total volume per sample) for 30 min on ice in the dark. Samples were washed twice with FACS buffer and resuspended in 200 μL 1X FoxP3 Fixation/Permeabilization Buffer (eBioscience) for 30 min in the dark. Each sample was then washed twice with 1X Permeabilization Buffer (eBioscience) and stained with intracellular antibodies in 1X Permeabilization Buffer supplemented with 10 μL Brilliant Stain Buffer Plus (100 μL total volume per sample) for 30 min in the dark. Samples were washed once with 1X Permeabilization Buffer, once with FACS buffer, resuspended in 200 μL FACS buffer and filtered through 70 um filter-cap FACS tubes, and acquired on a 5-laser (UV-V-B-YG-R) Cytek Aurora spectral flow cytometer (Cytek). UltraComp eBeads (Thermo Fisher) were used for single stain compensation controls. Fluorescence-minus-one controls were prepared for intracellular cytokine and transcription factor stains. All analysis of flow cytometry data was performed in FlowJo v10.8 (BD). For quality control, PBMC samples with <10% live cells or <2000 total live cells were excluded from final analyses. The flow staining strategy and exemplary gating are shown in Supplementary Fig. S6. The antibody panel is in Supplementary Table S2.

## Statistical analysis

Sample size calculation for this trial was based on the previously reported primary endpoint of MPR. DFS and PFS were estimated using the Kaplan-Meier method, and where appropriate, survival estimates were compared by the log rank test. All enrolled patients were randomized and considered assessable for survival analysis. All $p$-values are two-sided, with statistical significance evaluated at the 0.05 alpha level. Two-sided Student's $t$-test or Mann-Whitney test were used for the statistical analyses of ad-hoc RNAseq data. Benjamini-Hochberg correction was used to test for statistically significant differential gene expression. As no individual gene achieved a difference in expression following Benjamini-Hochberg correction, for exploratory studies, the 300 genes with the smallest non-adjusted $p$-values for differential expression were used for Gene Ontology pathway analyses. Analyses were performed in Prism GraphPad software (version 9.02), R (version 4.0.2 (Copyright [C] 2020; The R Foundation for Statistical Computing, Vienna, Austria)) and SAS Version 9.4 (SAS Institute, Inc., Cary, North Carolina).

For high-dimensional visualization of T cell flow cytometry data, PBMC samples were down-sampled to 10,000 CD3+ events and concatenated into a single sample. Pairwise controlled manifold approximation (PaCMAP) was performed on the concatenated sample in FlowJo using the PaCMAP plugin[43]. Unsupervised clustering of PaCMAP populations was done in FlowJo using the FlowSOM plugin[44]. Following initial QC exclusion, 104 (of the 126 total) PBMC samples were used to generate the CD4 and CD8 T cell PaCMAPs. In downstream analysis giving rise to figures, 4 additional samples that did not meet QC were excluded. Comparisons of T cell population frequencies were done using the Mann-Whitney test.

## Reporting summary

Further information on research design is available in the Nature Portfolio Reporting Summary linked to this article.

## Data availability

Deidentified patient data related to outcomes reported in this manuscript are available in the Source Data File. The RNAseq data generated in this study have been deposited in the GEO database under accession code GSE248378

The study protocol is available in the Supplementary Information file. The remaining data are available within the article, Supplementary Information or Source Data file. Source data are provided with this paper.

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

## Acknowledgements

This trial was funded by AstraZeneca, which provided durvalumab. The first author (N.K.A.) had full access to all the data in the study and had final responsibility for the decision to submit for publication. The funding source had no role in study design, conduct, data collection, data analysis or the writing of this report. Correlative studies were funded by the Neuberger Berman Lung Cancer Research Center of Weill Cornell Medicine. Generous support was also provided by the Yoram Cohen and Family research fund, the Victoria Moran-Furman and Jay Furman research fund, and the Patricia Donnington research fund. N.K.A., O.E., V.M., A.C.B and T.E.M. are supported in part by 5UH3CA244697. N.K.A. and T.E.M. are supported in part by Department of Defense grant W81XWH2110309. N.K.A. and V.M. are supported in part by National Cancer Institute grant 1 R01 CA271545-01A1. B.I. is supported by National Institute of Health grants, R37CA258829, R01CA280414, R01CA266446, U54CA274506, and additional by the Pershing Square Sohn Cancer Research Alliance Award, the Burroughs Wellcome Fund Career Award for Medical Scientists; a Tara Miller Melanoma Research Alliance Young Investigator Award; the Louis V. Gerstner, Jr. Scholars Program; and the V Foundation Scholars Award.

## Author contributions

N.K.A., B.I., and T.E.M. designed this study and interpreted the data. C.S. and A.N. collected the clinical data. A.C.B. generated and interpreted

the pathology and mutational data. P.C. and A.N. analyzed the survival data. T.E.M., O.E., B.B., and V.M. generated and interpreted the RNA sequence data. S.C.F. and N.J.S. designed the radiation schema. Z.H.W., A.A., J.M., P.H., C.A., L.C., and M.R. generated the flow cytometry and high-plex immunophenotyping. N.K.A., Z.H.W., B.I. and T.E.M. drafted and edited the final manuscript. J.L.P., B.E.L. and A.S. revised and edited the manuscript. N.K.A., Z.H.W., B.I., and T.E.M. verified the underlying data.

## Competing interests

N.K.A. reports stock options from TMRW, Angiocrine Bioscience, and View Point Medical; research advisory committee for AstraZeneca. J.L.P. reports leadershi1 and stock options from TMRW, Angiocrine Bioscience and View Point Medical. B.E.L. reports personal fees from AstraZeneca. A.S. reports personal fees from AstraZeneca, Blueprint Medicines, Genentech, Medtronic and Takeda. S.C.F. grants from Bristol Myers Squibb, Varian, Merck, Eisai, Elililly, Janssen and Regeneron; and personal fees from Accuray, AstraZeneca, Bayer, Bristol Myers Squibb, Eisai, Elekta, EMD Serano/Merck, GlaxoSmithKline, Janssen, MedImmune, Merck US, Regeneron, Varian, and ViewRay. B.l. is a consultant for or received honoraria from Volastra Therapeutics, Johnson & Johnson/Janssen, Novartis, Eisai, AstraZeneca and Merck, and has received research funding from Columbia University from Agenus, Alkermes, Arcus Biosciences, Checkmate Pharmaceuticals, Compugen, lmmunocore, and Synthekine. The other authors declare no competing interests.

## Additional information

Nasser K. Altorki [1,11] ✉, Zachary H. Walsh[2,11], Johannes C. Melms [2], Jeffery L. Port[1], Benjamin E. Lee[1], Abu Nasar[1], Cathy Spinelli[1], Lindsay Caprio[2], Meri Rogava[2], Patricia Ho [2], Paul J. Christos[3], Ashish Saxena[4], Olivier Elemento [5], Bhavneet Bhinder[5], Casey Ager[2], Amit Dipak Amin [2], Nicholas J. Sanfilippo[6], Vivek Mittal [1], Alain C. Borczuk [7], Silvia C. Formenti [6], Benjamin Izar [2,8,9,12] ✉ & Timothy E. McGraw [10,12] ✉

[1]Weill Cornell Medicine, Department of Cardiothoracic Surgery, New York, New York, USA. [2]Department of Medicine, Division of Hematology and Oncology, Columbia University Irving Medical Center, Vagelos College of Physicians & Surgeons, New York, New York, USA. [3]Department of Population Health Sciences, Weill Cornell Medicine, New York, New York, USA. [4]Weill Cornell Medicine, Division of Hematology and Oncology, New York, New York, USA. [5]Weill Cornell Medicine, Caryl and Israel Englander Institute for Precision Medicine, Institute for Computational Biomedicine, Department of Physiology and Biophysics, New York, New York, USA. [6]Weill Cornell Medicine, Department of Radiation Oncology, New York, New York, USA. [7]Department of Pathology, Northwell Health, Greenvale, New York, New York, USA. [8]Deparmtent of Systems Biology, Program for Mathematical Genomics, Columbia University, New York, New York, USA. [9]Columbia Center for Translational Immunology, New York, New York, USA. [10]Weill Cornell Medicine, Department of Biochemistry, New York, New York, USA. [11]These authors contributed equally: Nasser K. Altorki, Zachary H. Walsh. [12]These authors jointly supervised this work: Benjamin Izar, Timothy E. McGraw. ✉e-mail: nkaltork@med.cornell.edu; bi2175@cumc.columbia.edu; temcgraw@med.cornell.edu

