## [Peer Review File · Nature Communications]

Neoadjuvant durvalumab plus radiation versus durvalumab alone in stages I-III non-small cell lung cancer: survival outcomes and molecular correlates of a randomized phase II trialEditorial Note: This manuscript has been previously reviewed at another journal that is not operating a transparent peer review scheme. This document only contains reviewer comments and rebuttal letters for versions considered at *Nature Communications*.

REVIEWERS' COMMENTS:

Reviewer #1 (Remarks to the Author):

Thank the authors for revising the manuscript.

Another minor issue with the revised version of the manuscript” : in the Figure S1, please check the 2 boxes with MPR... something are missing or not shown up appropriately.

Reviewer #3 (Remarks to the Author):

Thank you for the opportunity to review this manuscript from Altorki et al. The authors have responded to my comments and the manuscript is significantly improved. I have a few additional minor comments.

The manuscript states that adjuvant chemotherapy was administered for 4 cycles. However, I don't believe it mentions how long adjuvant durvalumab was given.

There are some minor grammatical errors throughout the manuscript that should be corrected. For instance, in lines 386-387, it reads "High levels of post-treatment... is correlated with...". I believe this should be "High levels are correlated" or "High level is correlated", etc.

The discussion of the results of 3 year PFS in lines 442-445 is problematic. It seems hard to know whether a 10% difference in PFS in a small study with a P-value of 0.2 is "clinically meaningful." It may be worth softening this a bit.

Reviewer #4 (Remarks to the Author):

Thank you for the opportunity to review this manuscript as a new reviewer in commentary of the response to the prior reviewer comments. I believe the authors have made a reasonable effort to respond to the reviewer comments, although it is still very unclear in the abstract as to which treatment arm the results are pertaining to.

The response to the reviewers have raised additional potential queries around the reporting and design. The conclusion that there is no difference in outcome from a DFS and OS perspective is clouded by the underpowered nature of the comparison; despite 52 patients at risk there is clear separation on the curves between ARMS 1 and 2 in figures 2a-2c for survival.

The driving correlative outcomes are focused on CD103+ TRM CD4 and CD8 cells found in circulating in the periphery, and the association with MPR and response. Coupled with a lack of functional analysis and limited supportive translational biomarkers associated with outcome despite a relatively in-depth analysis, the new insights provided by this study are limited.

Reviewer #1

Thank the authors for revising the manuscript.

Another minor issue with the revised version of the manuscript” : in Figure S1, please check the 2 boxes with MPR... something are missing or not shown up appropriately.

We thank the reviewer for the positive evaluation of our work. We have clarified the minor issue noted by the reviewer in Figure S1.

Reviewer #3

Thank you for the opportunity to review this manuscript from Altorki et al. The authors have responded to my comments and the manuscript is significantly improved. I have a few additional minor comments.

We thank the reviewer for the positive evaluation of our work.

The manuscript states that adjuvant chemotherapy was administered for 4 cycles. However, I don't believe it mentions how long adjuvant durvalumab was given.

We apologize for the lack of clarity. We have now clarified that patients in the monotherapy arm received a median of 4.5 cycles of adjuvant durvalumab and that patients in the dual therapy arm received a median of 9.5 cycles of adjuvant durvalumab. The response is highlighted in the text.

There are some minor grammatical errors throughout the manuscript that should be corrected. For instance, in lines 386-387, it reads "High levels of post-treatment... is correlated with...". I believe this should be "High levels are correlated" or "High level is correlated", etc.

We have corrected all grammatical errors throughout the manuscript including the one cited by the reviewer (highlighted in text).

The discussion of the results of 3-year PFS in lines 442-445 is problematic. It seems hard to know whether a 10% difference in PFS in a small study with a P-value of 0.2 is "clinically meaningful." It may be worth softening this a bit.

We agree with the reviewer and have now deleted the phrase (clinically meaningful).

Reviewer #4 (Remarks to the Author):

Thank you for the opportunity to review this manuscript as a new reviewer in commentary of the response to the prior reviewer's comments. I believe the authors have made a reasonable effort to respond to the reviewer's comments, although it is still very unclear in the abstract as to which treatment arm the results are pertaining to.

We thank the reviewer for the overall positive evaluation of our revision. We apologize for the lack of clarity in the abstract. We have updated the abstract to emphasize more clearly that the results of the correlative studies are based on the analysis of samples obtained from patients in both treatment arms. The T cell results, as shown in Figures 3 and S2, are stratified by treatment arm. The gene expression data are based on the analysis of samples from patients with and without recurrence regardless of treatment arm.

The response to the reviewers have raised additional potential queries around the reporting and design. The conclusion that there is no difference in outcome from a DFS and OS perspective is clouded by the underpowered nature of the comparison; despite 52 patients at risk there is clear separation on the curves between ARMS 1 and 2 in figures 2a-2c for survival.

The reviewer raises an important point. We have reported the results of the study using the pre-specified levels of significance. It is important to note that there seems to be a trend towards improved survival in patients treated with durvalumab + SBRT compared to patients treated with durvalumab alone (shown in indicated Figures highlighted by the reviewer). We have modified the text in the discussion to emphasize the Reviewer's point (Highlighted). We would like to emphasize that this study, to our knowledge, is the first reported trial to date testing the combination of immunomodulatory doses of SBRT (8 Gyx3) plus IO in the neo-adjuvant setting for the treatment of early-stage NSCLC. The result of this pilot phase II trial is the basis for approval and funding of a multicenter randomized trial comparing SBRT+IO with the current standard of care of neoadjuvant chemotherapy +IO, the current standard of care. This new trial will be activated in the next several months. Thus the currently presented study is a landmark study that has already informed the field in an important manner.

The driving correlative outcomes are focused on CD103+ TRM CD4 and CD8 cells found in circulating in the periphery, and the association with MPR and response. Coupled with a lack of functional analysis and limited supportive translational biomarkers associated with outcome despite a relatively in-depth analysis, the new insights provided by this study are limited.

We thank the reviewer for raising this point. While emerging evidence suggests an important role for tumor-infiltrating T cells with tissue-residency memory (TRM) features in predicting response to IO based therapies, including another study since the review of our revision (<https://pubmed.ncbi.nlm.nih.gov/37683037/>), these are not easily procured in a sequential manner (e.g., before treatment, on-treatment, and at the time of drug resistance). Thus, evaluating the role of these CD103+ T cells in circulation offers an important opportunity for biomarker discovery, and our study provides the most comprehensive evidence for a role of these cells isolated from circulation. We agree that further validation in external cohorts is needed, and functional understanding of these cells is critical, and ought to be studied in appropriate models of TRM biology, as we have demonstrated in prior work (<https://www.sciencedirect.com/science/article/pii/S1535610822001672>). However, we strongly believe that this should not distract from the prospective opportunities this initial discovery provides, and our plan is to test the role of these cells in the clinical trial noted in response to this reviewer above. We also emphasized the need and opportunity that such a prospective trial has for addressing these questions.

REVIEWERS' COMMENTS

Reviewer #1 (Remarks to the Author):

Thank you for revising the Figure S1.

Reviewer #3 (Remarks to the Author):

The authors had addressed my comments. Thank you for the opportunity to review this manuscript.

Response to Reviewers

Reviewer 1

Thank you for revising the Figure S1.

Response: Thank you for reviewing our work

Reviewer 2

The authors had addressed my comments. Thank you for the opportunity to review this manuscript.

Response: We thank you for your supportive comments.